# Stigma and efficacy beliefs regarding opioid use disorder treatment and naloxone in communities participating in the HEALing Communities Study intervention

Nicky Lewis[1]*, Barry Eggleston[2], Redonna K. Chandler[3], Dawn Goddard-Eckrich[4], Jamie E. Luster[5], Dacia D. Beard[6], Emma Rodgers[4], Rouba Chahine[2], Philip M. Westgate[1], Shoshana N. Benjamin[4], JaNae Holloway[2], Thomas Clarke[3], R. Craig Lefebvre[2], Michael D. Stein[6], Donald W. Helme[1], Jennifer Reynolds[7], Sharon L. Walsh[1], Darcy Freedman[5], Nabila El-Bassel[4], Kara Stephens[7], Anita Silwal[1], Michelle Lofwall[1], Janet E. Childerhose[5], Hilary L. Surratt[1], Brooke N. Crockett[5], Amy L. Farmer[5], James L. David[4], Laura Fanucchi[1], Judy Harness[5], Ben Wilburn[7], Kelli Bursey[7], Kristin Mattson[7], Sarah Mann[5], Rebecca D. Jackson[5], Aimee Shadwick[8], Katherine Calver[9], Deborah Chassler[9], Jennifer Kimball[9], Nancy Regan[9], Jeffrey H. Samet[9], Rachel Sword-Cruz[9], Michael D. Slater[5]

1 University of Kentucky, Lexington, Kentucky, United States of America, 2 RTI International, Research Triangle Park, North Carolina, United States of America, 3 National Institute on Drug Abuse, Bethesda, Maryland, United States of America, 4 Columbia University, New York, North Carolina, United States of America, 5 The Ohio State University, Columbus, Ohio, United States of America, 6 Boston University, Boston, Massachusetts, United States of America, 7 Oak Ridge Associated Universities, Oak Ridge, Tennessee, United States of America, 8 RecoveryOhio, Columbus, Ohio, United States of America, 9 Boston Medical Center, Boston, Massachusetts, United States of America

* nicky.lewis@uky.edu

**Data Availability Statement:** The Steering Committee governs the Healing Communities

## Abstract

### Background

The HEALing Communities Study (HCS) included health campaigns as part of a community-engaged intervention to reduce opioid-related overdose deaths in 67 highly impacted communities across Kentucky, Massachusetts, New York, and Ohio. Five campaigns were developed with community input to provide information on opioid use disorder (OUD) and overdose prevention, reduce stigma, and build demand for evidence-based practices (EBPs). An evaluation examined the recognition of campaign messages about naloxone and whether stigma and efficacy beliefs regarding OUD treatment and naloxone changed in HCS intervention communities.

### Methods

Data were collected through surveys offered on Facebook/Instagram to members of communities participating in the HCS intervention and wait-list control communities.

### Results

Participants in HCS intervention communities reported a reduction in stigma regarding OUD and increased efficacy beliefs regarding naloxone associated with recognition of campaign

Study (HCS). Steering Committee membership consists of senior leadership at funders (NIDA, SAMHSA), the Research Sites (RS), and the Data Coordinating Center (DCC). Data release is governed by 20 Data Use Agreements (DUAs) involving the DCC with specific restrictions on data sharing by various state agencies and data owners. 1. The RSs and state agencies and cabinets are in the process of updating the DUAs to allow for data archival with our approved data masking protocols and detailed plans to lower the risk of deductive disclosure given the size of the smallest participating rural communities. 2. The data sharing plan will comply with the NIH HEAL Initiative® ClinicalTrials.gov Public Access and Data Sharing Policy, the NIH Data Sharing Policy, the NIH Policy on Dissemination of NIH-Funded Clinical Trial Information, and the NIH Clinical Trial Registration and Results Information Submission rule, and governing HCS data use agreements. 3. As such the HCS has created a specific data sharing plan that adheres to these legal requirements and restrictions. We plan to share allowable data in ICPSR's data repository (ICPSR Data Excellence Research Impact (umich.edu)) by March 2025. Additionally, HCS data will be searchable via NIH's HEAL data platform (healdata. org/landing). We plan to share data by March 2025 given the confidentiality, restrictions, and data governance enacted to protect communities, individuals, and our promises to data owners. We plan by March 2025 to make HEALing Communities Study methods, data, and results available to the public. The data sharing plan will comply with the NIH HEAL Initiative® ClinicalTrials. gov Public Access and Data Sharing Policy, the NIH Data Sharing Policy, the NIH Policy on Dissemination of NIH-Funded Clinical Trial Information, and the NIH Clinical Trial Registration and Results Information Submission rule, and governing HCS data use agreements. In the interim, all data inquiries can be sent to the DCC: HealingCommunities@rti.org.

**Funding:** All authors were supported via funding. Employees of the sponsors played a role in the study design, data collection and analysis, decision to publish, and preparation of the manuscript. https://www.nih.gov/ https://www.samhsa.gov/ Funding: This research was supported by the National Institutes of Health (NIH) and the Substance Abuse and Mental Health Services Administration through the NIH HEAL (Helping to End Addiction Long-termSM) Initiative under award numbers UM1DA049394, UM1DA049406, UM1DA049412, UM1DA049415, UM1DA049417 (ClinicalTrials.gov Identifier: NCT04111939). This study protocol (Pro00038088) was approved by

messages. However, this finding is cautiously interpreted as there was no clear evidence for recognition differences between the treatment/control conditions.

## Conclusion

Study findings indicate associations between campaign message recognition and positive outcomes. Results also highlight possible challenges concerning evaluations of social media campaigns using conventional evaluation techniques.

## Trial registration

ClinicalTrials.gov NCT04111939.

## Background

The U.S. continues to experience high rates of fatal drug overdoses; in 2022, nearly 105,000 individuals experienced fatal overdose [1]. Most are attributable to illicitly manufactured fentanyl and fentanyl analogues. One potential intervention to combat this epidemic is the use of health campaigns to reduce stigma around opioid use disorder (OUD) and increase support for evidence-based practices (EBPs) [2]. Most existing health campaigns addressing opioid overdose have not provided evidence of their effectiveness [3]. Health campaign evaluations are necessary to determine their effectiveness and to ensure wise use of community resources [4].

### Health campaigns' influence on knowledge, attitudes, and behavior

The influence of health campaigns on behavior depends largely on the context of the overall health intervention [5]; for example, whether the intervention has existing public support, public belief the intended behavior change is effective and feasibly achieved, and cultural acceptability. Other important elements include who the campaign focuses on and the content of the messages themselves.

A review of health communication and social marketing literature has generated EBPs that can guide the design of campaigns regarding opioid overdose [6]. Among these practices are the following: behavior change as an explicit goal; using formative research in design and planning; focusing on homogeneous populations; using multiple dissemination channels; increasing the frequency of conversations about specific health issues in social networks; prompting policy discussions that lead to effective policy changes; and ensuring the availability and access to services and resources to promote behavior change. These practices can increase public belief that the intended behavior change resulting from campaigns is achievable.

### The HEALing Communities Study (HCS)

The HCS is a multi-site, parallel-group, cluster randomized wait-list comparison trial testing the impact of the Communities That HEAL intervention (CTH) on opioid-related overdose deaths and other outcomes in 67 communities across four states (Kentucky, Massachusetts, New York, and Ohio). Communities were randomized to either receive the CTH first (Wave 1; $n = 34$) or serve as the wait-list control group (Wave 2; $n = 33$) to receive the CTH after the HCS comparison period was completed [7]. The CTH intervention includes three components: (a) a process for community coalition-driven decision making around the deployment/

Advarra Inc., the HEALing Communities Study single Institutional Review Board. The content is solely the responsibility of the authors and does not necessarily represent the official views of the National Institutes of Health, the Substance Abuse and Mental Health Services Administration or the NIH HEAL Initiative SM. Dr. Chandler participated in this paper consistent with her role as a NIH Science Officer.

**Competing interests:** I have read the journal's policy and the authors of this manuscript have the following competing interests: Michelle Lofwall has served as a scientific consultant for Berkshire Biomedical, Braeburn, Journey Colab, and Titan Pharmaceuticals in the last three years. This does not alter our adherence to all PLOS ONE policies on sharing data and materials.

expansion of EBPs guided by a data-driven approach to reduce opioid overdose deaths [8], (b) a set of EBPs with demonstrated efficacy and effectiveness in reducing overdoses and treating OUD [9], and (c) health campaigns to drive demand for EBPs, reduce stigma toward OUD treatment, and inform people about availability of OUD-related services [2]. Priority target groups for the campaigns included community leaders, people at risk for an overdose or with OUD, and family and friends of those at risk. Community-specific dissemination strategies were developed with local coalitions to reach these priority groups. One community initially randomized to receive the intervention decided not to participate in the study and withdrew before the CTH intervention began.

Over 27 months (April 1, 2020—June 30, 2022), five campaigns were sequentially launched in the Wave 1 (W1) intervention communities in the following order: overdose education and naloxone distribution (OEND); stigma related to medications for OUD (MOUD); MOUD Awareness; MOUD Treatment Retention; and Community Choice (i.e., communities chose to repeat a prior campaign). HCS community coalitions each selected Communications Champions who worked with research staff to develop community-specific content and disseminate campaign materials, using distribution plans. There were over 1,400 materials created for the first campaign (OEND; Campaign 1) that included print, billboards, and news stories, in addition to social media advertising. Campaign messaging had a consistent appearance and branding across the four states.

Recruitment for a Campaign Evaluation Questionnaire (CEQ) was conducted through social media (e.g., Facebook/Instagram) at regular intervals. The purpose of the CEQ was to assess campaign recognition and attitudes in W1 communities and less frequently in the Wave (W2) wait-list control communities. Social media served as a recruitment tool to assess recognition and impact of the campaigns in HCS communities.

### Research questions

This study examines two primary research questions: 1) did W1 community members accurately report recognition of (i.e., 'having seen') naloxone advertisements from the HCS campaigns compared to W2 community members that were in the control condition (i.e., did not have HCS campaigns underway), and 2) did Wave 1 community members show evidence of a reduction in individual-level stigma regarding OUD and improved efficacy beliefs regarding OUD treatment and naloxone?

### Methods

This study was designed as a longitudinal comparison of communities within the context of a cluster randomized trial. Of the 67 communities, 34 were randomized to W1, and 33 were randomized to W2 (i.e., wait-list control); W2 received the CTH intervention after the main trial evaluation period was over. One community withdrew immediately after randomization and was excluded from further analysis (i.e., was not treated as randomized). This report focuses on the per protocol population of the remaining 66 communities. All procedures were approved by the single Institutional Review Board Advarra Inc. (Pro00037850 for pilot test; Pro00038088 for final guide).

The CEQ, designed by HCS researchers, assessed specific components of stigma regarding OUD treatment and awareness and acceptability of naloxone and MOUD treatment. Recruitment and administration of the cross sectional CEQ survey occurred in W1 communities for approximately 5–6 weeks at each of six time points (i.e., CEQ 1–6), starting at baseline (March 2020; prior to Campaign 1) and then after each campaign (see Table 1 for the CEQ timeline). The CEQ was administered to W2 at baseline and twice thereafter in June—July 2021 and

**Table 1. Participant characteristics stratified by iteration of Campaign Evaluation Questionnaire (CEQ)[1].**

| Characteristic, statistic | Iteration | | | | | | Overall |
|---|---|---|---|---|---|---|---|
| | CEQ1 | CEQ2 | CEQ3 | CEQ4 | CEQ5 | CEQ6 | |
| | 3/30/20-4/22/20 | 9/2/20-9/30/20 | 1/13/21-2/18/21 | 6/14/21-7/21/21 | 11/8/21-12/17/21 | 7/1/22-8/11/22 | |
| **Number of Respondents** | 1,178 | 909 | 726 | 1,518 | 676 | 1,733 | 6,740 |
| **Number of Communities Represented** | 63 | 33 | 33 | 66 | 33 | 66 | 66 |
| **Intervention, n(%)** | | | | | | | |
| Wave 1 | 600 (50.9%) | 909 (100.0%) | 726 (100.0%) | 830 (54.7%) | 676 (100.0%) | 900 (51.9%) | 4,641 (68.9%) |
| Wave 2 | 578 (49.1%) | NA | NA | 688 (45.3%) | NA | 833 (48.1%) | 2,099 (31.1%) |
| **Research Site, n(%)** | | | | | | | |
| Kentucky | 370 (31.4%) | 124 (13.6%) | 97 (13.4%) | 392 (25.8%) | 109 (16.1%) | 449 (25.9%) | 1,541 (22.9%) |
| Massachusetts | 184 (15.6%) | 120 (13.2%) | 104 (14.3%) | 235 (15.5%) | 130 (19.2%) | 249 (14.4%) | 1,022 (15.2%) |
| New York | 229 (19.4%) | 246 (27.1%) | 237 (32.6%) | 393 (25.9%) | 180 (26.6%) | 497 (28.7%) | 1,782 (26.4%) |
| Ohio | 395 (33.5%) | 429 (46.1%) | 288 (39.7%) | 498 (32.8%) | 257 (38.0%) | 538 (31.0%) | 2,395 (35.5%) |
| **Geographic Location, n(%)** | | | | | | | |
| Rural | 191 (16.2%) | 200 (22.0%) | 186 (25.6%) | 445 (29.3%) | 189 (28.0%) | 475 (27.4%) | 1,686 (25.0%) |
| Urban | 987 (83.8%) | 709 (78.0%) | 540 (74.4%) | 1,073 (70.7%) | 487 (72.0%) | 1,258 (72.6%) | 5,054 (75.0%) |
| **Age, n(%)** | | | | | | | |
| 18–34 Years | 427 (36.4%) | 217 (25.3%) | 161 (23.7%) | 266 (20.0%) | 108 (17.5%) | 391 (22.8%) | 1,570 (24.6%) |
| 35–49 Years | 316 (27.0%) | 221 (25.7%) | 187 (27.5%) | 308 (23.2%) | 164 (26.5%) | 476 (27.8%) | 1,672 (26.2%) |
| 50–64 Years | 310 (26.5%) | 295 (34.3%) | 218 (32.1%) | 478 (36.0%) | 225 (36.4%) | 535 (31.2%) | 2,061 (32.3%) |
| 65–74 Years | 104 (8.9%) | 109 (12.7%) | 102 (15.0%) | 218 (16.4%) | 100 (16.2%) | 248 (14.5%) | 881 (13.8%) |
| 75+ Years | 15 (1.3%) | 17 (2.0%) | 12 (1.8%) | 59 (4.4%) | 21 (3.4%) | 63 (3.7%) | 187 (2.9%) |
| **Race/Ethnicity, n(%)** | | | | | | | |
| Non-Hispanic White | 961 (83.6%) | 683 (81.2%) | 552 (84.1%) | 1,067 (83.2%) | 500 (82.0%) | 1,387 (83.0%) | 5,150 (82.9%) |
| Non-Hispanic Black | 79 (6.9%) | 75 (8.9%) | 49 (7.5%) | 88 (6.9%) | 47 (7.7%) | 115 (6.9%) | 453 (7.3%) |
| Non-Hispanic Other | 54 (4.7%) | 45 (5.4%) | 25 (3.8%) | 61 (4.8%) | 34 (5.6%) | 74 (4.4%) | 293 (4.7%) |
| Hispanic | 55 (4.8%) | 38 (4.5%) | 30 (4.6%) | 66 (5.1%) | 29 (4.8%) | 96 (5.7%) | 314 (5.1%) |
| **Gender, n(%)** | | | | | | | |
| Male | 297 (25.4%) | 179 (20.8%) | 137 (20.1%) | 324 (24.4%) | 131 (21.2%) | 350 (20.4%) | 1,418 (22.3%) |
| Female | 847 (72.5%) | 666 (77.4%) | 530 (77.9%) | 981 (74.0%) | 475 (76.9%) | 1,327 (77.2%) | 4,826 (75.7%) |
| Other | 25 (2.1%) | 15 (1.7%) | 13 (1.9%) | 21 (1.6%) | 12 (1.9%) | 41 (2.4%) | 127 (2.0%) |
| **Education, n(%)** | | | | | | | |
| <High School | 36 (3.1%) | 21 (2.4%) | 22 (3.2%) | 67 (5.1%) | 32 (5.2%) | 86 (5.0%) | 264 (4.2%) |
| High School Diploma | 184 (15.8%) | 149 (17.4%) | 117 (17.3%) | 272 (20.5%) | 144 (23.3%) | 400 (23.4%) | 1,266 (19.9%) |
| Some College/Associate | 466 (39.9%) | 344 (40.1%) | 280 (41.4%) | 536 (40.5%) | 254 (41.0%) | 703 (41.1%) | 2,583 (40.6%) |
| College Degree | 285 (24.4%) | 195 (22.7%) | 159 (23.5%) | 249 (18.8%) | 113 (18.3%) | 285 (16.7%) | 1,286 (20.2%) |
| Graduate/Professional Degree | 196 (16.8%) | 149 (17.4%) | 99 (14.6%) | 201 (15.2%) | 76 (12.3%) | 236 (13.8%) | 957 (15.1%) |
| **Personal Experience with Opioid Addiction/Opioid Use Disorder[2]** | | | | | | | |
| Mean (SD) | 1.7 (1.3) | 1.7 (1.3) | 1.7 (1.4) | 1.8 (1.4) | 2.0 (1.4) | 1.9 (1.4) | 1.8 (1.4) |
| Median (Q1, Q3) | 2.0 (0.0, 3.0) | 2.0 (0.0, 3.0) | 2.0 (0.0, 3.0) | 2.0 (0.0, 3.0) | 2.0 (1.0, 3.0) | 2.0 (0.0, 3.0) | 2.0 (0.0, 3.0) |
| **Individual-Level Stigma, n(%)** | | | | | | | |
| **If I had an opioid addiction/opioid use disorder I would not tell anyone.** | | | | | | | |
| Strongly Agree | 118 (10.1%) | 84 (9.6%) | 79 (11.2%) | 120 (8.7%) | 67 (10.5%) | 161 (9.4%) | 629 (9.7%) |
| Agree | 309 (26.3%) | 243 (27.8%) | 172 (24.4%) | 336 (24.5%) | 139 (21.9%) | 396 (23.2%) | 1,595 (24.7%) |
| Neither Agree Nor Disagree | 273 (23.3%) | 217 (24.9%) | 162 (23.0%) | 356 (25.9%) | 165 (25.9%) | 453 (26.6%) | 1,626 (25.2%) |
| Disagree | 301 (25.7%) | 215 (24.6%) | 182 (25.9%) | 390 (28.4%) | 155 (24.4%) | 444 (26.0%) | 1,687 (26.1%) |
| Strongly Disagree | 172 (14.7%) | 114 (13.1%) | 109 (15.5%) | 171 (12.5%) | 110 (17.3%) | 251 (14.7%) | 927 (14.3%) |

*(Continued)*

**Table 1.** (Continued)

| Characteristic, statistic | Iteration | | | | | | Overall |
|---|---|---|---|---|---|---|---|
| | CEQ1 | CEQ2 | CEQ3 | CEQ4 | CEQ5 | CEQ6 | |
| | 3/30/20-4/22/20 | 9/2/20-9/30/20 | 1/13/21-2/18/21 | 6/14/21-7/21/21 | 11/8/21-12/17/21 | 7/1/22-8/11/22 | |
| **Efficacy Beliefs, n(%)** | | | | | | | |
| **Once you have an opioid addiction/opioid use disorder, there's not much you can do about it.** | | | | | | | |
| Strongly Agree | 7 (0.6%) | 4 (0.5%) | 3 (0.4%) | 17 (1.2%) | 5 (0.8%) | 25 (1.5%) | 61 (0.9%) |
| Agree | 26 (2.2%) | 10 (1.1%) | 9 (1.3%) | 24 (1.7%) | 8 (1.3%) | 27 (1.6%) | 104 (1.6%) |
| Neither Agree Nor Disagree | 56 (4.8%) | 35 (4.0%) | 35 (5.0%) | 83 (6.0%) | 41 (6.4%) | 118 (6.9%) | 368 (5.7%) |
| Disagree | 349 (29.7%) | 302 (34.4%) | 231 (32.8%) | 446 (32.5%) | 194 (30.4%) | 500 (29.1%) | 2,022 (31.2%) |
| Strongly Disagree | 737 (62.7%) | 526 (60.0%) | 427 (60.6%) | 803 (58.5%) | 391 (61.2%) | 1,048 (61.0%) | 3,932 (60.6%) |
| **I would be willing to carry naloxone when out in public.** | | | | | | | |
| Strongly Agree | 378 (32.3%) | 305 (34.9%) | 244 (34.8%) | 489 (36.0%) | 255 (40.1%) | 720 (42.0%) | 2,391 (37.1%) |
| Agree | 391 (33.4%) | 282 (32.3%) | 216 (30.8%) | 418 (30.7%) | 183 (28.8%) | 499 (29.1%) | 1,989 (30.8%) |
| Neither Agree Nor Disagree | 221 (18.9%) | 159 (18.2%) | 134 (19.1%) | 262 (19.3%) | 120 (18.9%) | 289 (16.9%) | 1,185 (18.4%) |
| Disagree | 132 (11.3%) | 79 (9.0%) | 63 (9.0%) | 127 (9.3%) | 37 (5.8%) | 131 (7.6%) | 569 (8.8%) |
| Strongly Disagree | 47 (4.0%) | 49 (5.6%) | 44 (6.3%) | 64 (4.7%) | 41 (6.4%) | 74 (4.3%) | 319 (4.9%) |
| **Baseline Opioid Overdose Death Rate[3]** | | | | | | | |
| Mean (SD) | 41.2 (14.3) | 38.3 (18.7) | 37.6 (20.8) | 39.0 (17.3) | 39.5 (20.8) | 39.3 (19.2) | 39.3 (18.3) |
| Median (Q1, Q3) | 45.5 | 38.9 | 34.5 | 42.0 | 37.3 | 38.9 | 42.0 |
| | (31.6, 49.1) | (21.8, 49.3) | (21.6, 49.3) | (25.4, 49.1) | (21.8, 49.3) | (25.4, 49.3) | (25.4, 49.3) |

[1]The number of communities participating in CEQs 1, 4, and 6 were larger than CEQs 2, 3, and 5 because only W1 communities were targeted in CEQs 2, 3, and 5.

[2]Sum of four questions where 1 = Yes and 0 = No:

1) I have had personal issues with opioid addiction/opioid use disorder.

2) A relative has had personal issues with opioid addiction/opioid use disorder.

3) A close friend has had personal issues with opioid addiction/opioid use disorder.

4) Someone I know personally has had issues with opioid addiction/opioid use disorder.

[3]Community-level rate per 100,000 residents ages 18+

June—August 2022, during which W2 communities had no exposure to the HCS campaigns. Our goal was to obtain $\leq 20$ surveys per community at each time point to achieve 80% power based on a two-sided test and a 5% significance level. Recruitment within a community was stopped at $n = 20$ respondents for each CEQ to avoid oversampling of more populated (urban) areas.

Potential CEQ survey participants were recruited as a convenience sample via a series of Facebook/Instagram advertisements that targeted zip codes corresponding to the HCS communities. Community residents $\geq 18$ years who resided in one of the 66 HCS communities were eligible to participate in data collection. Individuals who authorized Facebook/Instagram to collect certain information also had to report having a legitimate Facebook/Instagram account. They were then directed to a brief screening instrument to validate their place of residence (zip code), age (date-of-birth), and email address. Email addresses were entered on a separate form to preclude linkage to data.

Participants deemed eligible for the survey were routed to the CEQ survey instrument hosted on REDCap, where they were provided an electronic informed consent form to electronically sign. For those who agreed to participate, online consent was obtained. Upon

completion of the CEQ, each participant was offered the opportunity to enter a drawing to win a $100 Amazon electronic gift card. At the close of the CEQ, one electronic gift card was distributed to one winner in each of the 66 communities.

The CEQ collected demographic information from respondents. This information included race/ethnicity (coded as non-Hispanic White, non-Hispanic Black, Hispanic, or other); gender (male, female, different identity); age (18–34, 35–49, 50–64; 65–74, 75 and over); and education (high school or less, some college/associate degree, bachelor's degree, graduate degree).

## Outcome variables

**Message recognition measurement.**   Each state submitted an HCS advertisement from Campaign 1 used in their communities to test recognition in the CEQ. The advertisements submitted by each site were selected because they were widely disseminated in their respective communities. The naloxone advertisements appeared in CEQs 2–6. Respondents were asked how often they had seen it (1 = *Definitely Seen 5 or More Times*, 2 = *Definitely Seen 3 or 4 Times*, 3 = *Definitely Seen Once or Twice*, 4 = *Maybe Seen*, 5 = *Never Seen*, see [10]). To control for inaccurate self-report or false recognition among W1 participants, these findings were compared to W2 results.

**Individual-level stigma regarding OUD.**   Individual-level stigma regarding OUD was measured using a single item adapted from Griffiths et al.'s (2004) [11] research on stigmatizing attitudes toward depression. This item ($M$ = 3.11, $SD$ = 1.21) was answered on a 5-point Likert scale from 1 (*Strongly Agree*) to 5 (*Strongly Disagree*) and stated, "If I had an opioid addiction/opioid use disorder, I would not tell anyone." Higher response scores indicated less individual-level stigma.

**Efficacy beliefs regarding OUD treatment and naloxone.**   Two separate items were used to assess efficacy beliefs (defined as belief that an intended behavior change is both achievable and useful) regarding OUD treatment and naloxone. The first ($M$ = 4.49, $SD$ = 0.74) was adapted from Saunders et al.'s (2013) [12] questionnaire on health beliefs used a 5-point Likert scale from 1 (*Strongly Agree*) to 5 (*Strongly Disagree*). The item stated, "Once you have an opioid addiction/opioid use disorder, there's not much you can do about it." Higher response scores suggested more belief in efficacy regarding OUD treatment. A second item ($M$ = 2.13, $SD$ = 1.16) rated on the same 5-point Likert scale stated, "I would be willing to carry naloxone (more commonly known as Narcan®), the medication that can reverse an opioid overdose, when out in public" [13]. Higher response scores suggested less belief in efficacy regarding naloxone.

## Statistical analysis

Linear mixed models with robust, small sample corrected empirical standard error estimates were used for inference on the data collected [14]. Models examined differences between W1 and W2 residents on naloxone advertisement recognition and examined interaction effects involving time. Models also examined the relationship between advertisement recognition and outcomes of stigma/efficacy using CEQ1 through CEQ6 data from W1, where outcomes of stigma/efficacy were modeled separately. Finally, models examining the effect of message comprehension/impact on the relationship between advertisement recognition and outcomes of stigma/efficacy were fit using all W1 CEQ data.

All models included respondent demographics (including race/ethnicity), respondent opioid use experience-related variables, respondent community baseline opioid death rate, and state as covariates, and community as a random effect. Except for analyses considering the effect of comprehension/impact, analyses first assessed interactions involving time. If those

interaction analyses were insignificant, then effects were estimated without time. If analyses resulted in a significant time interaction, then effects were estimated at each post-baseline time point. For analyses of the effect of comprehension/impact, the three-way interaction between comprehension/impact, naloxone advertisement comprehension and time were first estimated. If the three-way interaction was not significant, then the two-way interaction was assessed. Effect estimates were least squares means extracted from fitted models.

The Benjamini-Hochberg (1995) [15] adjustment was used for multiple comparisons of two-way interactions in the analysis of effects involving naloxone advertisement recognition on stigma/efficacy as well as three and two-way interactions involving message comprehension/impact. Listwise deletion accounted for missing data. Analyses were conducted using proc GLIMMIX in SAS v9.4 [16].

### Covariates

Comprehension of the campaign messages themselves was run as a covariate using three items adapted from Sutfin et al.'s (2019) [17] research on tobacco health campaigns. These items addressed overall comprehension of the messages and included: "This message grabbed my attention;" "This message is easy for me to understand;" and "This message has a picture and text that match." Answered on a 5-point Likert scale from 1 (*Strongly Disagree*) to 5 (*Strongly Agree*), the three items were summed and averaged to create a message evaluation scale ($M = 11.6$, $SD = 2.32$) that demonstrated good reliability (Cronbach's $\alpha = 0.80$). A higher score indicated better comprehension.

Perceived message impact was also run as a covariate with five items created based on the goals of the HCS campaigns. Items included: "This message makes me want to carry naloxone when out in public;" "This message makes me want to learn more about medication for opioid use disorder (MOUD);" "This message recommends medication for opioid use disorder (MOUD);" "This message would encourage me to seek help if I had an opioid addiction/opioid use disorder;" and "This message would encourage me to seek help for a loved one if they had an opioid addiction/opioid use disorder." These items were also answered on a 5-point Likert scale from 1 (*Strongly Disagree*) to 5 (*Strongly Agree*). After summing the items, a scale of perceived message impact was created ($M = 16.84$, $SD = 4.18$), which had strong internal consistency (Cronbach's $\alpha = 0.86$). Higher response scores indicated greater message impact.

## Results

### Baseline characteristics

Participants' characteristics for the CEQs are summarized in Table 1. There were comparable numbers of participants across the states and waves as intended. The most frequently reported age group for participants was 50–64 years. In all CEQs, the majority of participants identified as white, female, and with some college or an associate degree. Participants tended to indicate 'yes' to at least one of four items related to personal experience with opioid addiction/opioid use disorder (see Table 1).

### W1 and W2 levels of HCS campaign recognition

CEQ1 (baseline), CEQ4, and CEQ6 data collected from W1 and W2 were compared. More than 65% of participants in both waves indicated they had never seen an HCS naloxone advertisement, with "maybe seen" as the second most common response (see Table 2). The comparative analysis revealed naloxone advertisement recognition was not significantly different by

**Table 2. Observed response frequencies by wave and time point.**

| CEQ | Wave | Response to Naloxone Advertisement Recognition Item | | | | |
|---|---|---|---|---|---|---|
| | | *Definitely Seen* *5+ times* | *Definitely Seen* *3 or 4 times* | *Definitely Seen* *1 or 2 times* | *Maybe Seen* | *Never Seen* |
| 1 | 1 | 16 | 8 | 36 | 97 | 423 |
| | | *2.8%* | *1.4%* | *6.2%* | *16.7%* | *72.9%* |
| | 2 | 8 | 11 | 33 | 87 | 422 |
| | | *1.4%* | *2.0%* | *5.9%* | *15.5%* | *75.2%* |
| 4 | 1 | 15 | 20 | 79 | 99 | 523 |
| | | *2.0%* | *2.7%* | *10.7%* | *13.5%* | *71.1%* |
| | 2 | 22 | 13 | 49 | 90 | 437 |
| | | *3.6%* | *2.1%* | *8.0%* | *14.7%* | *71.5%* |
| 6 | 1 | 25 | 26 | 100 | 158 | 581 |
| | | *2.8%* | *2.9%* | *11.2%* | *17.8%* | *65.3%* |
| | 2 | 25 | 38 | 74 | 110 | 581 |
| | | *3.0%* | *4.6%* | *8.9%* | *13.3%* | *70.2%* |

wave after averaging across time (Adjusted Wave Effect = -0.04; 95% CI [-0.14, 0.06], $p = 0.35$) or by rural/urban status ($p = 0.83$).

## HCS campaign recognition on individual-level stigma and efficacy beliefs regarding OUD treatment and naloxone

Examination of W1 CEQ data revealed a significant change over time for two outcomes (see Table 3). There was a positive relationship with recognition of naloxone advertisements, "If I had an opioid addiction/opioid use disorder I would not tell anyone" (individual-level stigma

**Table 3. Statistical analysis of relationship between naloxone advertisement recognition and one measure of individual-level stigma and two measures of efficacy beliefs regarding OUD treatment and naloxone.**

| Outcome | HCS Advertisement Recognition by Time Interaction | Estimated Average Slope Across Time |
|---|---|---|
| | FDR adj. p-value | Est. (95% CI) |
| If I had an opioid addiction/opioid use disorder I would not tell anyone.* | 0.62 | -0.063 (-0.119, -0.007) ** |
| Once you have an opioid addiction/opioid use disorder there's not much you can do about it.* | 0.62 | 0.034 (-0.004, 0.071) |
| I would be willing to carry Naloxone (more commonly known as Narcan®), the medication that can reverse an opioid overdose, when out in public.*** | 0.08 | 0.107 (0.081, 0.134) **** |

\* Negative estimate implies increased recognition of naloxone advertisement is associated with increased disagreement with the statement.

\*\* This estimated relationship implies the more recognition of naloxone advertisement; the more likely people disagree with the statement. In terms of stigma, this estimate is interpreted as: for each unit increase in recognition of naloxone advertisement estimated individual-level stigma reduced by 0.06; 95% CI [0.01, 0.12].

\*\*\* Positive estimate implies increased recognition of naloxone advertisement is associated with increased agreement with the statement.

\*\*\*\* This estimated relationship implies the more recognition of naloxone advertisement; the more likely people agree with the statement.

regarding OUD) and "willingness to carry naloxone" (efficacy beliefs regarding naloxone). For each unit increase in recognition of the naloxone advertisement over time, estimated individual-level stigma was reduced by 0.06; 95% CI [0.01, 0.12] and willingness to carry naloxone increased by 0.11; 95% CI [0.08, 0.13]. It is important to note that unique cohorts were recruited for each CEQ, thus changes observed in W1 do not represent within-subject changes over time. Moreover, there was no evidence that message comprehension or impact (i.e., the covariates) influenced the relationship between reported naloxone advertisement recognition and the outcome variables over time.

## Discussion

### Main finding of this study

Among survey respondents from communities participating in the HCS, there was no statistically significant difference in naloxone advertisement recognition between W1 and W2 communities. However, W1 respondents who reported advertisement recognition over time revealed reduced stigma ratings regarding OUD and increased willingness to carry naloxone as a function of advertisement recognition. Although this effect was small, modest effect sizes for mass media health campaigns implemented in large populations can instigate substantial prosocial change [5].

The lack of significant differences between W1 and W2 respondents for recognition of the HCS naloxone advertisements could be due to insufficient exposure to the advertisements among W1 respondents [18]. However, the finding that nearly a third of W2 respondents indicated that they had seen an HCS naloxone advertisement also suggests that there may have been 1) misidentification from exposure to other communication campaigns or advertisements related to naloxone, 2) an acquiescence or agreement bias to the images and questions, 3) possible interaction with HCS campaigns in W1 communities (e.g., traveling to nearby W1 counties), or 4) that they were exposed to other HCS messages through public service announcements, community events, and earned media such as news coverage, which were not assessed here.

### Limitations of this study

First, we recruited our sample via Facebook/Instagram, resulting in a convenience sample biased toward social media users. It is worthwhile to note, however, that each CEQ recruited a unique cohort of individuals reaching a large population ($n$ = 4,641 in W1). Second, HCS campaigns were targeted toward the HCS priority groups (community leaders, people at risk, and family and friends of those at risk), while the CEQ was distributed toward a more general audience. Third, the CEQ sample was not diverse, especially regarding gender and race, limiting generalizability to other populations or settings. Finally, W2 data were collected at only three of the six time points used in the treatment communities, preventing us from conducting a useful group comparison for the analysis of stigma and efficacy beliefs.

### What this study adds

Although participants in W1 did report positive changes in stigma regarding OUD and efficacy toward carrying naloxone, the lack of difference between self-reported message recognition in treatment versus control communities reduces confidence in claims of campaign impact. Digital campaign impressions were reasonably high as HCS process data revealed over 15 million social media impressions for the OEND advertisements. What might then account

for the lack of meaningful recognition of campaign messages relative to controls in the treatment communities as indicated by these data?

Use of longitudinal data collection with message recognition measures is a validated approach to assessing public health communication interventions [5, 19]. Our results suggest that this method, typically used with television and radio advertising, in-school posters, and other hard-to-ignore media, may be problematic for evaluating social media advertising. Such advertisements appear in social media feeds and users are well-accustomed to ignoring those that are not of enough interest for a close read and possible click-through. Therefore, such messages may leave little or no effect in the absence of viewer interest [20].

If most social media advertising has little or no impact on users who are unmotivated regarding the topic, this would have important implications. Social media advertisements may be good candidates for reaching targeted, motivated audiences and directing them to websites, in-depth information, and resource referrals. Evaluations using techniques such as website tracking studies should focus on identifying such effects. However, reliance on social media advertising may pose more difficulties in terms of trying to reach a general population with attempts to influence attitudes and behavior. Furthermore, message recognition-based evaluation designs may be a poor fit to assessing such social media efforts, unless closely linked to targeted populations for whom the message is relevant. Future public health communication campaign research might further examine these possibilities to help guide evaluation efforts and to make the best possible use of social media advertising channels.

## Acknowledgments

We wish to acknowledge the participation of the HEALing Communities Study communities, community coalitions, study team staff and faculty, community advisory boards, partner agencies, and state government officials who collaborated with us on this study. In addition, we would like to thank the community members who volunteered to be included in the campaign materials that shared their images and stories.

## Author Contributions

**Conceptualization:** Nicky Lewis, Redonna K. Chandler, Dacia D. Beard, Emma Rodgers, R. Craig Lefebvre, Michael D. Stein, Donald W. Helme, Jennifer Reynolds, Sharon L. Walsh, Nabila El-Bassel, Kara Stephens, Laura Fanucchi, Ben Wilburn, Kelli Bursey, Kristin Mattson, Deborah Chassler, Jeffrey H. Samet, Michael D. Slater.

**Data curation:** Nicky Lewis, Barry Eggleston, Rouba Chahine, JaNae Holloway, R. Craig Lefebvre.

**Formal analysis:** Rouba Chahine.

**Funding acquisition:** Sharon L. Walsh, Nabila El-Bassel, Rebecca D. Jackson, Jeffrey H. Samet.

**Investigation:** Nicky Lewis, Dawn Goddard-Eckrich, Jamie E. Luster, Dacia D. Beard, Emma Rodgers, R. Craig Lefebvre, Sharon L. Walsh, Nabila El-Bassel, Hilary L. Surratt, Jeffrey H. Samet.

**Methodology:** Nicky Lewis, Barry Eggleston, Dawn Goddard-Eckrich, R. Craig Lefebvre, Sharon L. Walsh, Laura Fanucchi, Michael D. Slater.

**Project administration:** Nicky Lewis, Dawn Goddard-Eckrich, Jamie E. Luster, Dacia D. Beard, Emma Rodgers, R. Craig Lefebvre, Michael D. Stein, Donald W. Helme, Sharon L.

Walsh, Nabila El-Bassel, James L. David, Sarah Mann, Rebecca D. Jackson, Jeffrey H. Samet, Michael D. Slater.

**Resources:** Nicky Lewis, Dawn Goddard-Eckrich, Dacia D. Beard, Emma Rodgers, R. Craig Lefebvre, Sharon L. Walsh, James L. David, Aimee Shadwick.

**Software:** Barry Eggleston, Rouba Chahine.

**Supervision:** Nicky Lewis, Redonna K. Chandler, Dawn Goddard-Eckrich, Philip M. Westgate, R. Craig Lefebvre, Donald W. Helme, Sharon L. Walsh, Nabila El-Bassel, James L. David, Rebecca D. Jackson, Jeffrey H. Samet, Michael D. Slater.

**Validation:** Nicky Lewis, Redonna K. Chandler, Rouba Chahine, JaNae Holloway, R. Craig Lefebvre, Sharon L. Walsh.

**Visualization:** Nicky Lewis, Redonna K. Chandler, Philip M. Westgate.

**Writing – original draft:** Nicky Lewis, Redonna K. Chandler, Dawn Goddard-Eckrich, Jamie E. Luster, Dacia D. Beard, Emma Rodgers, Philip M. Westgate, R. Craig Lefebvre, Jennifer Reynolds, Sharon L. Walsh, Kara Stephens, Ben Wilburn, Kelli Bursey, Kristin Mattson, Michael D. Slater.

**Writing – review & editing:** Nicky Lewis, Barry Eggleston, Redonna K. Chandler, Dawn Goddard-Eckrich, Jamie E. Luster, Dacia D. Beard, Emma Rodgers, Rouba Chahine, Philip M. Westgate, Shoshana N. Benjamin, JaNae Holloway, Thomas Clarke, R. Craig Lefebvre, Michael D. Stein, Donald W. Helme, Jennifer Reynolds, Sharon L. Walsh, Darcy Freedman, Nabila El-Bassel, Kara Stephens, Anita Silwal, Michelle Lofwall, Janet E. Childerhose, Hilary L. Surratt, Brooke N. Crockett, Amy L. Farmer, James L. David, Laura Fanucchi, Judy Harness, Ben Wilburn, Kelli Bursey, Kristin Mattson, Aimee Shadwick, Katherine Calver, Deborah Chassler, Jennifer Kimball, Nancy Regan, Jeffrey H. Samet, Rachel Sword-Cruz, Michael D. Slater.

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
