## [Decision Letter · Decision Letter 0]

22 May 2024

PONE-D-24-07101How do stigma and efficacy beliefs regarding opioid use disorder treatment and naloxone change in communities participating in the HEALing Communities Study intervention?PLOS ONE

Dear Dr. Lewis,

Thank you for submitting your manuscript to PLOS ONE. After careful consideration, we feel that it has merit but does not fully meet PLOS ONE’s publication criteria as it currently stands. Therefore, we invite you to submit a revised version of the manuscript that addresses the points raised during the review process.

We look forward to receiving your revised manuscript.

Kind regards,

Sairah Hafeez Kamran, PhD

Academic Editor

PLOS ONE

Journal Requirements:

"I have read the journal's policy and the authors of this manuscript have the following competing interests: Michelle Lofwall has served as a scientific consultant for Berkshire Biomedical, Braeburn, Journey Colab, and Titan Pharmaceuticals in the last three years."

4. In this instance it seems there may be acceptable restrictions in place that prevent the public sharing of your minimal data. However, in line with our goal of ensuring long-term data availability to all interested researchers, PLOS’ Data Policy states that authors cannot be the sole named individuals responsible for ensuring data access (http://journals.plos.org/plosone/s/data-availability#loc-acceptable-data-sharing-methods).

Reviewers' comments:

Reviewer's Responses to Questions

**Comments to the Author**

1. Is the manuscript technically sound, and do the data support the conclusions?

Reviewer #1: Yes

Reviewer #2: Yes

2. Has the statistical analysis been performed appropriately and rigorously? 

Reviewer #1: Yes

Reviewer #2: Yes

3. Have the authors made all data underlying the findings in their manuscript fully available?

Reviewer #1: No

Reviewer #2: No

4. Is the manuscript presented in an intelligible fashion and written in standard English?

Reviewer #1: Yes

Reviewer #2: Yes

5. Review Comments to the Author

Reviewer #1: This is a very interesting study. Kindly see my minor comments below.

1. Why are there so many authors involved in this study?

2. In the methodology section, the study design used is not clearly indicated. Please amend this section to include information on the study design used.

3. Why did you use per-protocol analysis instead of intention-to-treat analysis in your study?

4. From lines 135–39, there are different time points for conducting interviews, i.e., the CEQ timeline, for the intervention group (W1) and comparison group (W2). How could these differences contribute to bias and impact the validity of your result?

5. In line 146, what do you mean by the statement “residents ≥ 18”? If you intend to indicate the inclusion criteria of your study participants as “residents ≥ 18 years," please amend it to reflect the proposed change.

6. In Line 183, the statement “Higher response scores suggested less belief in efficacy regarding OUD treatment” is not clear. The given 5-point Likert scale from 1 (strongly agree) to 5 (strongly disagree) for the item stated, “Once you have an opioid addiction or opioid use disorder, there’s not much you can do about it,” indicated that the higher response scores suggested a higher belief in efficacy regarding OUD treatment. Kindly provide explanations for this observation or amend it accordingly to be in line with the proposed 5-point Likert scale. A similar observation was seen in lines 184–87.

7. In the result section, why were only 63 communities included in Table 1 for baseline data (CEQ1) instead of 66 communities?

The overall quality of English is good, but language and grammar edits can further strengthen the manuscript.

Reviewer #2: The title of the article can be trimmed a little, it is 23 word titles can be reduced to 20 or less. The data is not made available but explanation is given as to why, which is understandable. The article will add to the body of knowledge.

6. PLOS authors have the option to publish the peer review history of their article (what does this mean?). If published, this will include your full peer review and any attached files.

Reviewer #1: No

Reviewer #2: **Yes: **Abdulrahman Ahmad

---

## [Author Response · Author response to Decision Letter 0]

29 Jul 2024

July 1, 2024

Dear Dr. Kamran,

Thank you for the opportunity to revise and resubmit our manuscript: “How do stigma and efficacy beliefs regarding opioid use disorder treatment and naloxone change in communities participating in the HEALing Communities Study intervention?,” now shortened to: “Stigma and efficacy beliefs regarding opioid use disorder treatment and naloxone in communities participating in the HEALing Communities Study intervention.” The reviewer comments were considered at length, and new versions of the manuscript have been submitted to the Author Center (one marked and one unmarked). Please see a point-by-point response to the reviewer comments below.

From Reviewer 1:

This is a very interesting study. Kindly see my minor comments below.

Thanks to Reviewer 1 for their support of our manuscript.

1. Why are there so many authors involved in this study?

This is a fair question. The HEALing Communities Study (HCS) is the largest implementation science study ever conducted in addiction research and included 67 communities across 4 states. Our large multi-site, interdisciplinary teams included faculty, staff, and community members from multiple domains (e.g., communication, community engagement, statistics, clinical trials, opioid use disorder treatment and overdose prevention prevention). Our commitment to team science, valuing site members and including contributions by community members, was a priority for this and all other HCS manuscripts. 

2. In the methodology section, the study design used is not clearly indicated. Please amend this section to include information on the study design used.

We added additional content to clarify the study’s design. The first sentence of the methods section (p. 6) now describes the study as a longitudinal comparison of communities within the context of a cluster randomized trial. We also included a sentence at the top of p. 6 explaining that recruitment via social media served as a tool to assess recognition and impact of the campaigns. In the middle of p. 7, we added an additional note that the CEQ study utilized a convenience sample. If there is additional information we should add about the HCS and the CEQ, we would be happy to do so.

3. Why did you use per-protocol analysis instead of intention-to-treat analysis in your study?

Per-protocol analysis was used instead of intention-to-treat analysis because one community was randomized but dropped out before the intervention began. This community was randomized to the intervention arm (W1) but never had exposure to any of the communication campaigns, or other aspects of the HCS intervention. A sentence has been added to the bottom of p. 6 explaining that one community withdrew immediately after randomization, was not treated as randomized, and excluded from further analysis. 

4. From lines 135–39, there are different time points for conducting interviews, i.e., the CEQ timeline, for the intervention group (W1) and comparison group (W2). How could these differences contribute to bias and impact the validity of your result?

Indeed, W2 only had data collection at certain time points compared to W1. However, the different time schedules for the CEQ between the waves should not cause any biases because in cases where analysis involved a comparison between W1 and W2, those analyses only included CEQ data from timepoints where they were collected from both Waves (CEQs 1, 4, and 6). Data from CEQs 2, 3, and 5 were only used in analysis involving Wave 1. 

5. In line 146, what do you mean by the statement “residents ≥ 18”? If you intend to indicate the inclusion criteria of your study participants as “residents ≥ 18 years," please amend it to reflect the proposed change.

Thanks to Reviewer 1 for catching this. This line has now been changed to add ‘years’ after 18. 

6. In Line 183, the statement “Higher response scores suggested less belief in efficacy regarding OUD treatment” is not clear. The given 5-point Likert scale from 1 (strongly agree) to 5 (strongly disagree) for the item stated, “Once you have an opioid addiction or opioid use disorder, there’s not much you can do about it,” indicated that the higher response scores suggested a higher belief in efficacy regarding OUD treatment. Kindly provide explanations for this observation or amend it accordingly to be in line with the proposed 5-point Likert scale. A similar observation was seen in lines 184–87.

We see where this wording could be confusing. The wording has been amended to read correctly at the bottom of p. 8 and at the top of p. 9. The original version of the manuscript included incorrect wording but had no effect on the interpretation of the analyses. 

7. In the result section, why were only 63 communities included in Table 1 for baseline data (CEQ1) instead of 66 communities?

The CEQ was a survey sent out via social media, so it is possible that no one from a particular community clicked on the link during a specific CEQ. As such, CEQ1 only had 63 communities represented by participants. Alternatively, CEQ4 had all 66 communities represented by participants.

The overall quality of English is good, but language and grammar edits can further strengthen the manuscript.

We have conducted an additional review of language and grammar to ease readability.

Our thanks again to Reviewer 1 for their thoughtful critique and comments on our manuscript.

From Reviewer 2:

The title of the article can be trimmed a little, it is 23 word titles can be reduced to 20 or less. The data is not made available but explanation is given as to why, which is understandable. This article will add to the body of knowledge.

We agreed with Reviewer 2 that the title could be shortened. It is now 20 words. Thanks also to Reviewer 2 for their positive feedback on our manuscript. 

We greatly appreciate the Editor and Reviewers for taking the time and effort to thoughtfully review our manuscript. Thank you again for considering this manuscript for publication in PLOS ONE.

---

## [Editor Report · Decision Letter 1]

5 Aug 2024

Stigma and efficacy beliefs regarding opioid use disorder treatment and naloxone in communities participating in the HEALing Communities Study intervention

PONE-D-24-07101R1

Dear Dr. Lewis,

We’re pleased to inform you that your manuscript has been judged scientifically suitable for publication and will be formally accepted for publication once it meets all outstanding technical requirements.

Kind regards,

Sairah Hafeez Kamran, PhD

Academic Editor

PLOS ONE

---

## [Editor Report · Acceptance letter]

8 Aug 2024

PONE-D-24-07101R1 

PLOS ONE

Dear Dr. Lewis, 

I'm pleased to inform you that your manuscript has been deemed suitable for publication in PLOS ONE. Congratulations! Your manuscript is now being handed over to our production team.

Kind regards, 

on behalf of

Dr. Sairah Hafeez Kamran 

Academic Editor

PLOS ONE